

# Construction of a high-density linkage map and detection of sex-specific markers in *Penaeus japonicus*

Yaqun Zhang[1],[*], Chuantao Zhang[2],[*], Na Yao[1], Jingxian Huang[2], Xiangshan Sun[2], Bingran Zhao[2] and Hengde Li[1]

[1] Chinese Academy of Fishery Sciences, Beijing, China
[2] Xiaying Enhancement and Experiment Station, Chinese Academy of Fishery Sciences, Weifang, Shandong, China
[*] These authors contributed equally to this work.

## ABSTRACT

*Penaeus japonicus* is one of the most important farmed shrimp species in many countries. Sexual dimorphism is observed in *P. japonicus*, in which females grow faster and larger than males; therefore, a unisexual female culture of *P. japonicus* could improve the efficiency of productivity. However, the genetic mechanisms underlying sex determination in *P. japonicus* are unclear. In this study, we constructed a high-density genetic linkage map of *P. japonicus* using genotyping-by-sequencing (GBS) technology in a full-sib family. The final map was 3,481.98 cM in length and contained 29,757 single nucleotide polymorphisms (SNPs). These SNPs were distributed on 41 sex-averaged linkage groups, with an average inter-marker distance of 0.123 cM. One haplotype, harboring five sex-specific SNPs, was detected in linkage group 1 (LG1), and its corresponding confidence interval ranged from 211.840 to 212.592 cM. Therefore, this high-density genetic linkage map will be informative for genome assembly and marker-assisted breeding, and the sex-linked SNPs will be helpful for further studies on molecular mechanisms of sex determination and unisexual culture of *P. japonicus* in the future.

Corresponding authors
Bingran Zhao, bran6888@163.com
Hengde Li, hengde.li@cafs.ac.cn

## INTRODUCTION

The kuruma shrimp, *Penaeus japonicus*, is considered to be one of the most economically important members of the family Penaeidae, and it is distributed along the east coast of South Africa and in Red Sea, Indian Ocean, Korea, Japan, China, Malaysia, Philippines, Indonesia, Fiji Island, and North Australia (*Hayashi, 1996*). The annual production of *P. japonicus* in China was approximately 50,000 tons from 2013 to 2019 (*Ministry of Agriculture & Rural Affairs fishery & Fishery Administration, 2013–2019*). Sexual dimorphism, in which females grow faster and achieve a larger size than males, occurs in *Penaeus* shrimps. Therefore, the growth superiority of female penaeid shrimp provides researchers an incentive to investigate the potential of producing and culturing all-female

shrimp populations. By removing slow-growing males, it is likely that the culturing of all-female shrimp could increase production and reduce the cost of farming. Therefore, the mechanism of sex determination in *Penaeus* has long been a question of great interest to researchers.

Genotyping-by-sequencing (GBS) is a method used to discover genome-wide high-throughput single nucleotide polymorphisms (SNPs) and perform genotyping studies simultaneously, and it functions by reducing genome complexity, relying on restriction enzymes and high-throughput sequencing technology (*Wallace & Mitchell, 2017*). Construction of genetic maps, which can obtain genomic and genetic variation information based on thousands of SNPs, is an important part of animal and plant molecular breeding and is of great significance for the rapid and scientific identification of molecular markers of target traits (*Huang, 2016*; *You, Shan & Shi, 2020*).

Simple sequence repeats (SSRs) are approximately 1% in most genomes of species and are considered to have no function; however, penaeid shrimp genomes have a high proportion of SSRs (>23%) (*Yuan et al., 2021*), which hinders genome assembly. Recently, the genome sequence of the Pacific white shrimp *L. vannamei* was reported to cover ~1.66 Gb with 25,596 protein-coding genes and a high proportion of SSRs. Genome sequence assembly provides insights into the genetic underpinnings of specific biological processes and valuable information for promoting crustacean aquaculture (*Zhang et al., 2019*). In addition to *L. vannamei*, genome sequencing and draft assembly of two economically important penaeid shrimps, *P. japonicus* and *P. monodon*, have also been reported, only at the scaffold level (*Yuan et al., 2018*). Recently, a chromosome-level assembly of the black tiger shrimp, *P. monodon*, was completed (*Uengwetwanit et al., 2021*). High-density genetic maps are essential and helpful for genome assembly at higher levels, comparative genomic analysis, and fine mapping of complex traits. To date, with the rapid development and application of high-throughput sequencing technology, high-density genetic maps of aquatic shrimps, including *Litopenaeus vannamei* (*Peng et al., 2020*; *Yu et al., 2015*; *Zhang et al., 2007*), *Penaeus monodon* (*Guo et al., 2019*; *Wilson et al., 2002*), and *Fenneropenaeus chinensis* (*Li et al., 2006*; *Wang et al., 2012*), have been completed for reduced-representation genome sequencing. Based on this, the genetic mechanisms underlying sex determination have also been explored. In *L. vannamei* (*Yu et al., 2017*), 11 significant SNPs (in high linkage disequilibrium) located on LG42 and 44 involved in sex determination were identified. Sex locus was detected in *P. monodon* and was speculated to be the same, based on sequence alignments in populations of Mozambique, India, and Hawaii (*Guo et al., 2019*). In 2013, the first genetic map of the kuruma prawn *P. japonicus* was constructed using AFLP markers (*Li et al., 2003*), and 217 markers were ordered into 43 linkage groups (1,780 cM) of the paternal map, while 125 markers were ordered into 31 linkage groups (1,026 cM) of the maternal map. In 2016, a higher-resolution genetic linkage map containing 9,289 SNP markers, spanning 3,610.90 cM and ordered into 41 linkage groups, was constructed using RAD technology. Growth-related quantitative trait locus (QTL) has also been identified in *P. japonicus*

(*Lu et al., 2016*). However, there are little published data on the molecular markers of sex determination in *P. japonicus*.

To explore the genetic basis of sex determination in *P. japonicus*, a genetic linkage map with higher resolution was constructed using GBS technology, and the sex-specific QTL was identified using chi-square test in this study, our results support the WZ/ZZ sex determination system.

## MATERIALS & METHODS

### Sample collection

The full-sib kuruma prawn family that was used for QTL mapping was an F2 population. In 2020, one full-sib family, including F1 parents ($n = 2$) and F2 offspring ($n = 200$) were randomly collected from the Xiaying Enhancement and Experiment Station, Chinese Academy of Fishery Sciences. Sex was determined by observing their sexual characteristics, the male shrimp has a male appendage on the inner edge of the second appendage, the female shrimp has a seminal vesicle located between the base of the fourth and fifth pairs of feet. One appendage for each prawn was sampled for DNA extraction.

### DNA extraction, library construction, and sequencing

Genomic DNA was extracted from each individual using TIANamp Marine Animals DNA Kit (TIANGEN, Beijing, China) and qualified using gel electrophoresis. DNA concentrations were measured on a Nanodrop and diluted to 50 ng/µL. GBS technology was used to construct sequencing libraries (*Qi et al., 2018*). The DNA concentration of each GBS library was quantified using a Qubit™ dsDNA HS assay kit. Then libraries were equally pooled and sequenced on an Illumina Nova platform (paired-end 150 bp).

### Genotyping

The raw reads were first split by barcode using the module 'process_radtags' within the Stacks v2.1 (*Catchen et al., 2013*) (options: -r -renz_1 –adapter_mm 1), then forward reads were filtered using barcode and restriction enzyme sites; they were considered qualified if they simultaneously carried both the barcode and the *Pst*I restriction site. The restriction sites and all bases at the 3′ end with scores less than 20 were removed using the FASTX Toolkit v0.0.14 package (http://hannonlab.cshl.edu/fastx_toolkit/). The clean reads within each sample were clustered using the 'ustacks' module (options: -m 2 -M 1 -N 1) of the Stacks (*Catchen et al., 2013*). The representative tags were identified across samples with 'ASustacks', and tag reads were removed if they occurred in less than 50% of samples or had similarity of ≥98% between samples. Preprocessed reads were aligned to the *de novo* reference sequences using bowtie2 (v2.3.4.3) (*Langmead & Salzberg, 2012*) with default parameters, and the genotypes were called using GATK (v 3.8-1) (*McKenna et al., 2010*). To obtain robust results in subsequent analyses, the genotypes were filtered based on three criteria using vcftools (v0.1.13) (*Danecek et al., 2011*): loci with sequencing depth < 8, minor allele frequency (MAF) < 0.01, and call rate < 80% were removed.

**Table 1 Sex-specific primer sequences designed according to the genomic sequence containing the identified markers.**

| Primer name | Primer sequence (5′-3′) |
| --- | --- |
| contig55055-F | GCGCTGTGCAATATAACAGTCATGG |
| contig55055-R | GTGGAATTATGACAGGTTCTGGACC |
| contig42978-F | TTCGGCATATAGATGGATCC |
| contig42978-R | CACTTCAATGACTCGTTGTG |
| contig29802-F | AACAGATCTCAAGGCACTG |
| contig29802-R | GCAGAACCAATTATGAAGACG |
| contig23315-F | GGATGAGCTGGTACTTCAATCACG |
| contig23315-R | TCAGTGGCGTTTCTCTACCTGTAGG |

## Linkage map construction

Linkage map was constructed using Lep-MAP3, which can handle large number of markers (*Rastas, 2017*) and involved several steps as follows: calling parental genotypes using ParentCall2 module, followed by filtering markers based on high segregation distortion (options: -dataTolerance 0.0001 -MAFLimit 0.05 -missingLimit 0.2) using Filtering2 module. Thereafter, markers were assigned to linkage groups (LGs) by computing all pair-wise likelihood of odds (LOD) scores between markers using SeparateChromosomes2 module, and joining markers with LOD scores higher than the parameter of "usePhysical" as 1, in which LOD score of 12 was used as the threshold. Finally, the markers within each LG were ordered by maximizing the likelihood of the data for alternative orders using OrderMarkers2 module. Joinsignles2All module was not used because only four markers were not assigned to the existing LGs. A linkage map was drawn using the R package 'LinkageMapView' (*Ouellette et al., 2018*).

## Sex-specific marker mapping and verification

Missing genotype information was imputed through linkage disequilibrium with 20 closest neighboring markers as previously described by *Jiang & Li (2017)*. The final genotypic data (Table S1) in the analysis were used to calculate the genotype frequency of each SNP, and then a chi-square test was performed to determine the association between the genotypes and the sexes. A genome-wise significance threshold was set as $1.0 \times 10^{-8}$ of *P*-value. The corresponding confidence interval was calculated using the method described by *Li (2011)*. PCR primers (Table 1) were designed to detect the sex-specific markers according to the contigs containing the identified markers (Table S2). The PCR reaction conditions were as follows: initial denaturation at 95 °C for 3 min; 35 cycles at 95 °C for 20 s for denaturation, 60 °C for 20 s for annealing, and 72 °C for 30 s for extension, followed by a final extension at 72 °C for 5 min. The amplified products were separated using 1.5% agarose gel electrophoresis, and the purified products were sequenced. The sex-specific SNP markers were determined manually by the peaks of sanger sequencing of PCR products. Furthermore, more adult shrimps from other group were used to validate the SNP markers (38 females, 51 males).

## RESULTS

### Genotypes

Parents and 200 progenies generated 229.96 million clean reads, comprising approximately 312.77 Gb of sequencing data, with 49.28% GC content (Table S2). A total of 68,457 SNPs were detected and successfully genotyped using GATK pipeline. After quality control, 29,773 high-quality SNPs were obtained.

### Linkage map

Among the 29,773 SNPs, only four SNPs were not mapped to the genetic linkage map in the SeparateChromosomes2 step. In addition, two LGs comprising only three SNPs and nine SNPs, respectively, which are relatively too short for one chromosome, were excluded from the map. The final map consisted of 41 LGs, harboring 29,757 SNPs. The total map length was 3,481.982 cM, with an average inter-locus distance of 0.123 cM. The genetic length of LGs ranged from 54.491 (LG29) to 214.101 cM (LG1), with an average inter-locus distance of 0.098 cM (Table 2 and Fig. 1). The sex-averaged map information is presented in Table S3. The lengths of the maternal and paternal maps were 3,481.982 and 3,469.499 cM, respectively, and correspondingly, they ranged from 50.584 (LG36) to 214.675 cM (LG1) and from 52.74 (LG37) to 169.327 cM (LG1), with an average inter-locus distance of 0.125 and 0.123 cM, respectively (Table S4).

### Sex QTL and validation of sex associated markers

One cluster of five SNPs around position 212.34 cM on LG1 was associated with the sex of *P. japonicus* (Fig. 2), in which alleles formed two haplotypes, $H_1$: GCAGC, $H_2$: CTCAT, and the genotypes of females and males were $H_1H_2$ and $H_1H_1$, respectively (Fig. 3), indicating a WZ/ZZ sex determination system in *P. japonicus*. The QTL confidence interval ranged from 211.840 to 212.592 cM in LG1. The tag sequence containing sex-specific loci was aligned against NCBI *P. japonicus* genome database, "the retrotransposon: Penelope-like element" (NCBI ID: AB612264.1) was hit, then blastx was conducted using the NCBI AB612264.1 sequence against Nr database, and the putative reverse transcriptase (NCBI ID: BAM35674.1) was annotated with an E value of 0. Primers were designed for the five potential sex-specific markers according to the corresponding contigs, and validated in another population (38 females, 51 males). They were successfully amplified (Fig. 3), sequencing results demonstrated that females are all heterozygous and males are all homozygous for these five sex-specific markers, and the specificity of these five markers in the discrimination of the population were all 100%.

## DISCUSSION

Sexual dimorphism is observed in *P. japonicus*, in which females grow faster and larger than males; thus a unisexual female culture of *P. japonicus* could improve the efficiency of productivity. Therefore, the sex determination system and the genomic regions associated with sex in *P. japonicus* needs to be explored. This study identified a QTL region containing five markers that explains 100% of the phenotypic variation in sex. These

Table 2 Summary of the consensus linkage map in *Penaeus japonicus*.

| Linkage group | Number of markers | Estimated linkage group length of consensus map | Number of unique markers |
|---|---|---|---|
| 1 | 1,640 | 214.101 | 216 |
| 2 | 1,182 | 100.837 | 172 |
| 3 | 1,171 | 98.44 | 151 |
| 4 | 1,105 | 108.945 | 143 |
| 5 | 1,096 | 99.991 | 141 |
| 6 | 1,044 | 124.737 | 150 |
| 7 | 964 | 93.295 | 132 |
| 8 | 916 | 89.627 | 120 |
| 9 | 886 | 88.463 | 124 |
| 10 | 870 | 87.312 | 125 |
| 11 | 863 | 95.234 | 136 |
| 12 | 852 | 94.383 | 132 |
| 13 | 836 | 115.051 | 126 |
| 14 | 829 | 62.96 | 111 |
| 15 | 825 | 83.777 | 134 |
| 16 | 775 | 85.193 | 112 |
| 17 | 765 | 95.459 | 130 |
| 18 | 714 | 102.954 | 107 |
| 19 | 707 | 88.387 | 124 |
| 20 | 695 | 86.752 | 117 |
| 21 | 681 | 83.739 | 122 |
| 22 | 648 | 61.875 | 78 |
| 23 | 644 | 77.304 | 109 |
| 24 | 643 | 85.181 | 116 |
| 25 | 623 | 78.426 | 97 |
| 26 | 603 | 99.836 | 130 |
| 27 | 583 | 88.12 | 109 |
| 28 | 582 | 59.774 | 93 |
| 29 | 558 | 54.491 | 72 |
| 30 | 555 | 77.963 | 107 |
| 31 | 553 | 69.978 | 78 |
| 32 | 552 | 63.152 | 77 |
| 33 | 510 | 65.272 | 69 |
| 34 | 488 | 56.542 | 71 |
| 35 | 487 | 66.683 | 73 |
| 36 | 482 | 77.136 | 81 |
| 37 | 431 | 59.904 | 68 |
| 38 | 366 | 63.75 | 69 |
| 39 | 357 | 56.515 | 68 |
| 40 | 352 | 62.691 | 73 |
| 41 | 324 | 57.752 | 68 |
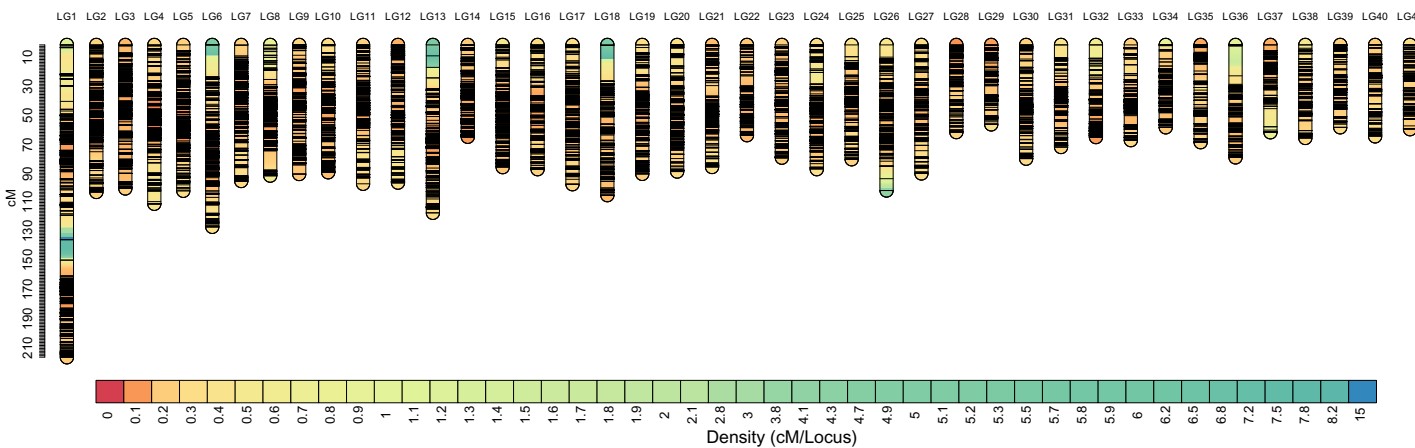

**Figure 1** **High-density sex-averaged genetic linkage map of *P. japonicus*.** The X-axis represents the linkage group, while the Y-axis represents the genetic position.

markers may be beneficial to identify underlying sex-determination genes, and also could be utilized to increase the proportion of females within the industry of kuruma prawn.

*Lu et al. (2017)* reported the genetic map of *P. japonicus* containing 9,289 SNP markers and spanning 3,610.90 cM with an average marker interval of 0.388 cM, and all the SNP markers were grouped into 41 LGs in the maps. In this study, the marker density of the constructed genetic map was three times higher than that reported previously, the final map of *P. japonicus* consisted of 29,757 SNPs that also clustered into 41 LGs; the average inter-locus distance was 0.123 cM; the higher resolution of the genetic map is beneficial to QTL mapping and genomic selection of growth, disease resistance and other complex traits in *P. japonicus*. A total of 27.14% scaffolds (4,943/18,210) of the assembled *P. japonicus* genome GCA_017312705.1 were mapped using contigs harboring 29,757 SNPs consisting of the final genetic map, indicating that the genetic map could provide a reference for chromosome-level assembly of the *P. japonicus* genome. Notably, karyotype analysis showed that *P. japonicus* has 43 chromosomes (*Xiang et al., 1991*), which is inconsistent to the 41 LGs obtained in this study. The corresponding number of LGs, which changed with the LOD limit, is shown in Fig. S1, the number of LGs with no less than 10 markers tended to be stable at 41 when the LOD limit was between 12 and 21. What needs to be explained is that the length of the two LGs containing 3 and 9 SNPs were only 0.503 cM and 4.382 cM, which was far below the length of the shortest LG (57.752 cM) in the final genetic map, together with that previous karyotype analysis showed that no extreme short chromosomes existed in *P. japonicus*, therefore, these two LGs were excluded from the final genetic map. We infer that chromosome rearrangements might occur in *P. japonicus*, leading to changes in chromosome numbers. Recently, chromosome rearrangements have been detected in the barred knifejaw *Oplegnathus fasciatus*, and a centric fusion of acrocentric chromosomes Ch8 and Ch10 should be responsible for the formation of the $X_1X_2Y$ system (*Xiao et al., 2020*). In addition, the difference in chromosomal numbers based on karyotyping and next generation sequencing

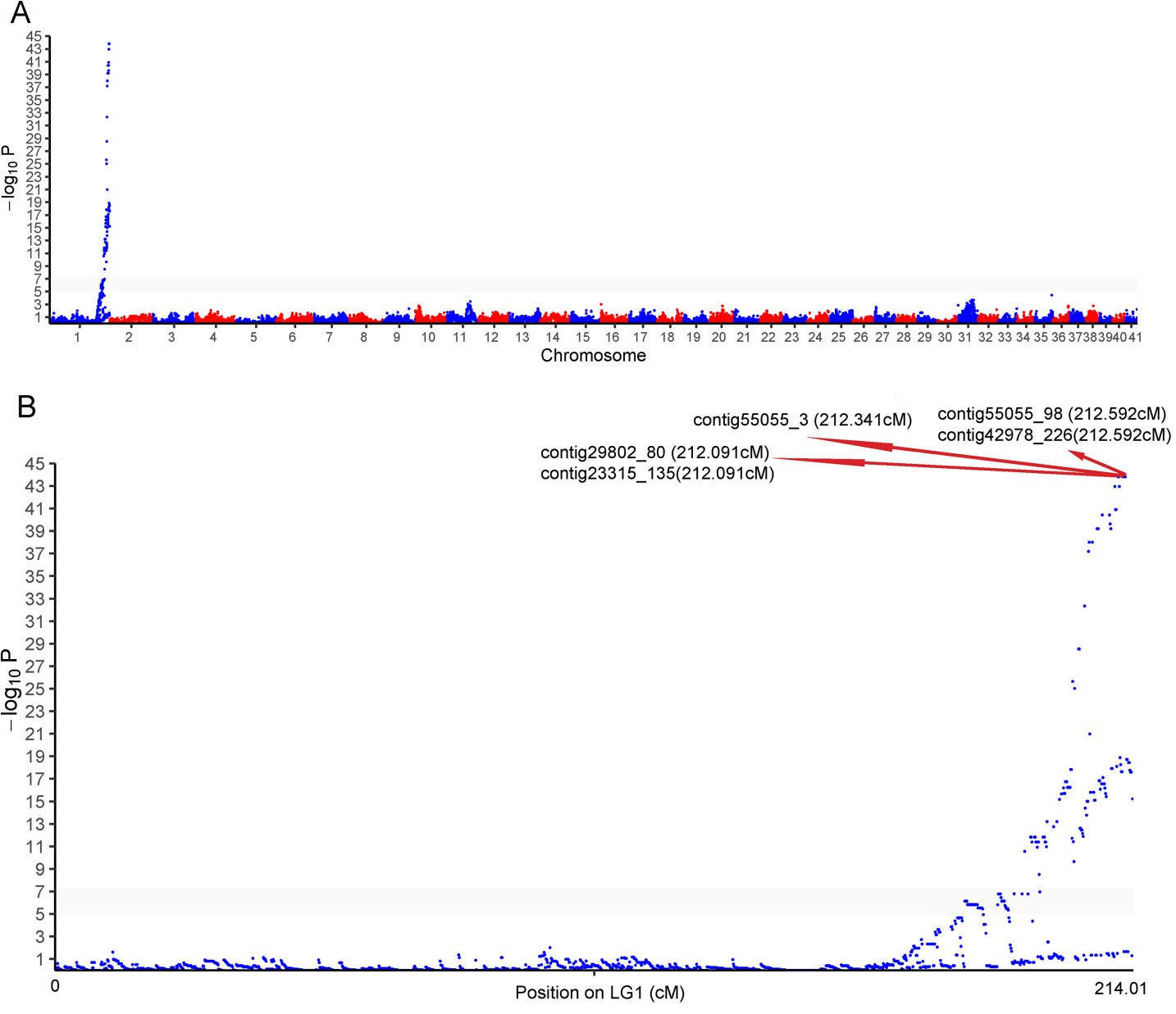

**Figure 2 Genome-wide manhattan plot associated to sex in *P. japonicus*.** (A) Manhattan plot of SNPs associated with sex, x-axis presents genomic coordinates along chromosome 1–41. The y-axis presents a negative logarithm of *P*-values. (B) Enlarged plot for LG1, the five sex-related SNPs were arrowed.

could be due to the misidentification of chromosome numbers, especially in species with large chromosome numbers such as crustacean species (*Waiho et al., 2021*).

Sex determination and differentiation processes has been among the most interesting topics in aquaculture. Sexual dimorphism is directly related to the economic benefits of some aquatic species, such as the crabs *Scylla paramamosain* and *Eriocheir sinensis*, and the females are preferred because of the yolk; the female Chinese tongue soles (*Cynoglossus semilaevis*) are more preferred owing to their faster growth rate and larger individual size than those of the males. Previous studies have identified sex-related QTLs and genes in

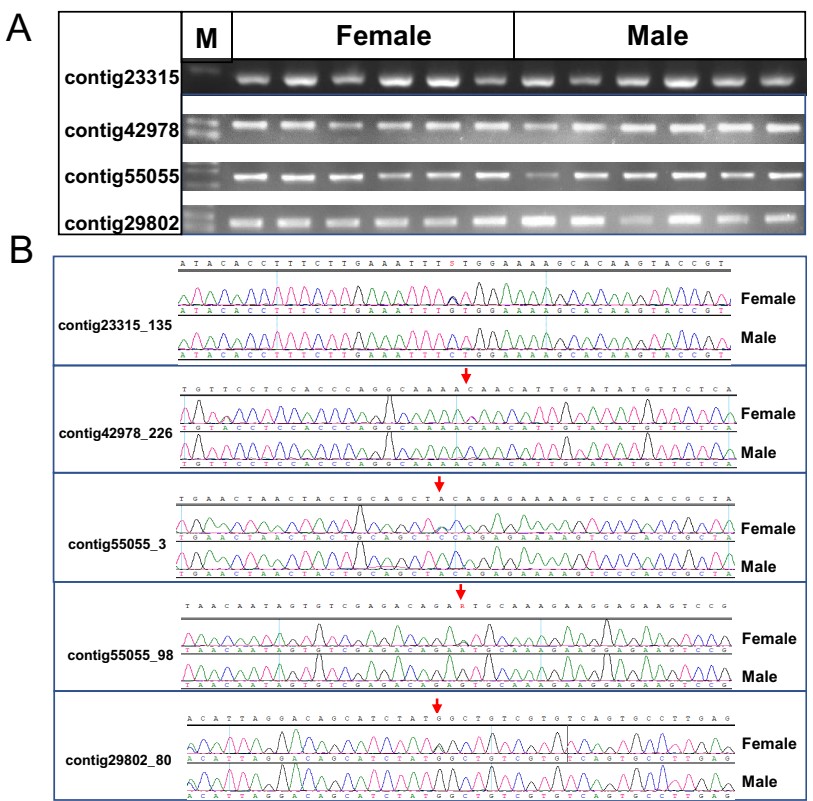

**Figure 3  Verification of the identified sex-related SNPs.** (A) Gel picture of the four contigs containing five sex-related SNPs. (B) Sanger sequencing showed differentiation of SNPs between females and males of *P. japonicus*, contig23315_135, C/G; contig42978_226, C/T; contig55055_3, C/A; contig55055_98, A/G; contig29802_80, C/T.               

fish (*Sun et al., 2017*; *Wei, Chen & Wang, 2019*), crabs (*Waiho et al., 2019*; *Shi et al., 2018*), and shrimps, including *L. vannamei* and *P. monodon* (*Jones et al., 2020*; *Wang et al., 2020*). In *L. vannamei*, sex-associated markers were identified on LG42.44 *via* mapping analysis, and the QTL region supports the ZW-WW chromosomal sex determination system; however, no direct sex determination or differentiation gene could be identified in this research (*Jones et al., 2020*). Four validated sex-linked SNPs on two sex-linked genes unigene0020898 and unigenen0020336 were identified by another research group, and the two genes might participate in sex determination and differentiation processes in *L. vannamei* (*Wang et al., 2020*). One sex locus was located (*Guo et al., 2019*; *Robinson et al., 2014*; *Staelens et al., 2008*), supporting that the sex of the black tiger shrimp is determined by a WZ/ZZ chromosomal system. However, studies on sex determination system of *P. japonicus* and sex-related markers are much limited. In this study, a sex-linked significant QTL including five SNPs was detected in LG1 of *P. japonicus*, which were heterozygous in all females but homozygous in all males, the segregation patterns of females:males for these five SNPs are 1:1, completely conforms to Mendelian separation law, also suggesting WZ/ZZ sex determination system in *P. japonicus*. One reverse transcriptase was annotated by this region. However, it has not been reported to be relevant to sex determination, further studies are needed to identify the genes involved in

sex-determining mechanisms in *Penaeus* species. As sexual dimorphism is observed in *P. japonicus* in which females grow faster and larger than males, the unisexual female culture could improve the productivity, however, it is difficult to distinguish the sex of *P. japonicus* at their early developmental stages. In this study, sex-specific primers were designed to amplify the region containing sex-related SNPs, and this PCR-based sex identification method were validated to identify the sex of *P. japonicus* successfully. Therefore, this sex QTL not only offers clues to explore the underlying molecular mechanism of sex determination and differentiation but could also be applied to sex identification and manipulation in the kuruma prawn industry, which leads to the increase of production in *P. japonicus* industry.

## CONCLUSIONS

GBS technology was applied to construct a high-density genetic linkage map for kuruma prawn *P. japonicus* in this study, this high-quality genetic linkage map will provide a reference for further genome assembly and genomic selection for important economic traits. Our results suggest a WZ/ZZ sex determination system in *P. japonicus*. The identified sex QTL did not only lay a research foundation for investigating the molecular mechanism of sex determination and differentiation of *P. japonicus*, but also provides theoretical support for possible unisexual breeding.

## ACKNOWLEDGEMENTS

We would like to thank Editage for English language editing.

### Funding

This work was supported by the Central Public-interest Scientific Institution Basal Research Fund [grant number CAFS: NO. 2021A004]; and Special Scientific Research Funds for Central Non-profit Institutes [grant number CAFS: 2020TD24]. The funders had no role in study design, data collection and analysis, decision to publish, or preparation of the manuscript.

### Grant Disclosures

The following grant information was disclosed by the authors:
Central Public-interest Scientific Institution Basal Research Fund: CAFS: NO. 2021A004.
Special Scientific Research Funds for Central Non-profit Institutes: CAFS: 2020TD24.

### Competing Interests

The authors declare that they have no competing interests.

### Author Contributions

- Yaqun Zhang analyzed the data, prepared figures and/or tables, authored or reviewed drafts of the paper, and approved the final draft.

- Chuantao Zhang performed the experiments, prepared figures and/or tables, and approved the final draft.
- Na Yao analyzed the data, authored or reviewed drafts of the paper, and approved the final draft.
- Jingxian Huang performed the experiments, prepared figures and/or tables, and approved the final draft.
- Xiangshan Sun performed the experiments, prepared figures and/or tables, and approved the final draft.
- Bingran Zhao conceived and designed the experiments, performed the experiments, authored or reviewed drafts of the paper, and approved the final draft.
- Hengde Li conceived and designed the experiments, analyzed the data, authored or reviewed drafts of the paper, and approved the final draft.

### Data Availability
Data is available at NCBI: PRJNA739820.

### Supplemental Information
Supplemental information for this article can be found online at http://dx.doi.org/10.7717/peerj.12390#supplemental-information.

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
