# Peer review of "Construction of a high-density linkage map and detection of sex-specific markers in Penaeus japonicus"

_PeerJ, doi:10.7717/peerj.12390_

## Round 0.1 · original submission · Major Revisions

I agree with the comments of all reviewers that although the results of this manuscript are novel and warrant publication, there are some major issues, particularly found in the Results and Discussion section, that need to be addressed before the manuscript is acceptable in PeerJ.

In addition to the reviewers' comments, do kindly address:

1. Line 85-87: The parents and offspring used are unclear. Do you mean F1 parents (n = 2) and F2 offspring (n = 200)? Also, what was the growth stage of the offspring during sample collection to allow for the accurate identification of sex? How was sex identified? Please detail these out.

2. Line 211: for crabs, I think you are referring to Waiho et al. 2019 (doi: 10.3389/fgene.2019.00298) instead of Shi et al. 2018. The former, not the latter, identified sex-related QTLs in Scylla paramamosain.

3. Kindly add another subfigure to Figure 2, detailing the position of the six sex-related SNPs and the corresponding QTL on LG1.

4. The authors should include in their results, a section on sex QTL instead of focusing directly on sex-specific markers.

5. The overall language of the manuscript needs further improvement. Kindly proofread the whole document.

6. Line 175-186 should be in the introduction section.

7. Line 193-204: The difference in chromosomal numbers based on karyotyping and next-generation sequencing could be due to chromosome rearrangement as noted by the authors, or simply due to misidentification, as seen in some species of brachyuran crabs (see review, doi: 10.1016/j.aquaculture.2021.736990)

8. Is it sufficient enough to postulate the ZW/ZZ sex determination system solely based on the difference in genotyping of SNP? Do include the segregation patterns and female:male recombination rate ratios as well to show the consistency of ZW/ZZ patterns.

9. The whole discussion section is too weak and needs to be re-written. The current discussion includes a lot of unnecessary information but does not discuss, in detail, the results of this study.

10. I agree with reviewer 2 that the sex-specific marker should be validated in more samples, preferably in a population with known pedigree information. The results of the validation (including a figure of the PCR validation gel picture) should be added to the Results section.

·

Basic reporting

no comment

Experimental design

no comment

Validity of the findings

no comment

Additional comments

Comments to the Author:

The article entitled “Construction of a high-density linkage map and detection of sex-specific markers in Penaeus japonicus” by Zhang et al. constructed a high-density genetic linkage map and identified sex QTL for kuruma prawn P. japonicus. This study not only provide a reference for further genome assembly and genomic selection for important economic traits, but also lay a research foundation for investigating the molecular mechanism of sex determination and differentiation of P. japonicus, which could provide theoretical support for unisexual breeding. The results obtained are beneficial both in theory and application. The tables are sufficiently organized, and the figures are well presented in the manuscript. Overall, the study deserves publication. However, there are still some issues needed to be corrected and clarified.

1. Line 29: “an average inter-marker distance of 0.117 cM”. However, in the text, you presented that “an average inter-locus distance of 0.123 cM” on Line 153. Please clarify which is right?

2. Line 49-52: This sentence is not related to this paper, please delete it or replace it by other contents.

3. Line 73: you used “,” in “9,289” but not in “3610.90”. Please unify them in the manuscript.

4. Line 88: “Sex was determined by observing their sexual characteristics”, this is intricate to readers who are not very professional. Please state how to distinguish the sex of P. japonicus. For example, you can explain them by describing, pictures of sexual characteristics, or the references.

5. Line 135-136, I am confused about this sentence “The feasibility of this pair of sex-specific primers was validated using PCR with samples from six females and six males”. I wonder which would you want to validate, primers or SNP markers? Besides, I think six males and six females are too few to validate the SNP markers. What’s more, where did these 12 individuals obtain, from the 200 offspring or from other groups? Please clarify.

6. Line 150-151: “two LGs comprising only three SNPs and nine SNPs, respectively, which are relatively too short for one chromosome, were excluded from the map”. Is it logical and reasonable to remove SNPs and delete chromosomes from the genetic map just because the SNP markers are scarce? Please give the explanation.

7. Line 160-163, you said that six SNPs on LG1 were obviously associated with the sex. However, you only presented one SNP marker in Figure 3. The sex-specific markers are very important for MAS breeding. Please show the detail information of these six SNPs, containing the genotypes of females and males and the picture of Sanger sequencing of every SNP marker.

8. Line 172-174, this sentence is repetitive and useless, please delete it.

9. Line 186-188, “9289 SNP markers” appeared twice in this sentence, please rewrite it.

10. Line 190, “average inter-locus distance was 0.12 cM”, and “0.123 cM” was presented in Line 153. Please unify them in the manuscript.

11. Line 212 and Line 215, “Jones et al., 2020b” and “Jones et al., 2020a” appeared, respectively. But I can’t find these two articles in the reference. Please clarify.

12. Line 218-220, you said “One sex locus was located, supporting that the sex of the black tiger shrimp is determined by a WZ-ZZ chromosomal system”. I wonder that this sex locus was identified by three articles? Why did you give three references in this sentence?

13. Line 220-222, please give the name of the species you described in this sentence.

14. The discussion section is illogical, chaotic and simple. I suggest the authors revising this section carefully.

Reviewer 2 ·

Basic reporting

no commont

Experimental design

no comment

Validity of the findings

no comment

Additional comments

The authors constructed a high-density genetic linkage map and develop a sex specific marker for Penaeus japonicus. This resources would benefit the basic research of sex determining mechanism and breeding in aquaculture. However, some issues must be explained more clear, especially the efficiency of the sex specific marker.
1.Line 31. delete “on LG1”
2.Line 41. please cite the book in references.
3.Line 64-65. please rephrase this sentence. The technology of AFLP is independent of GBS or RADseq.
4.Line 95-96. a library....the libraries
5.Line 99 rephrase this sentence. Two “using”.
6.Line 127. please rephrase this sentence.
7.Line 127 and 128. please briefly describe the method.
8.Line 136. are the six females and six males from the full-sib family? Please validate the sex specific marker in more population.
9.Line 149 vs line 124 “only four SNPs” vs “only two markers” are they inconsistent?
10.Line 160. what is the percent of phenotype that these marker explained?
11.Line 160. “obviously” is not suitable.
12.Line 162-163. How to explain the inconsistency of the genotypes between SNP and haplotype? Females are homozygous in haplotype genotypes while heterozygous in SNP genotypes.
13.Line 196. The choice of LOD is irresponsible. Actually, the lower limit of marker number of a group may vary from species to species. So the common feature would be the true linkage groups would contain relative more markers. Thus, the marker size distribution and the number of groups should be pay attention to at the same time. Usually, set the low limit of marker number infinity and observe the distribution of marker numbers with different LOD. When the best LOD chosen, the SNPs are expected to be assigned into stable number of linkage group. Thus, the discussion about chromosome rearrangement is not reasonable.
14.Table1. Please supplement the unique number of marker and change “Number of markers” to “Number of SNP”
15.Rephrase the legend of Figure 2

---

## Round 0.2 · Major Revisions

I congratulate the authors for their efforts in revising the manuscript. However, I agree with R2 that sex-linked SNP validation is necessary and on top of that, the authors could consider improving the LOD value to ensure that the number of linkage group and chromosome number (via karyotyping) is identical. Do bear in mind that when the authors shift the LOD value, please make the necessary amendments to other relevant sections and results too.

·

Basic reporting

no comment

Experimental design

no comment

Validity of the findings

no comment

Additional comments

1. Line 56: you deleted the sentence, but you did not delete the reference of “Thitamadee et al., 2016” in Line 462.

2. Different forms of “WZ/ZZ” were presented in the MS. For example, “WZ-ZZ” in Line 103, and “WZ/ZZ” in Line 207. Please unify them in the manuscript.

3. Line 175, please clarify the number of males and females in these 89 adult shrimps.

4. Line 200: change “3481.982” to “3,481.982”. Please check the MS carefully, and unify them.

5. Line 206: two “respectively” in this sentence.

6. Line 237, please add “marker” before “density”.

7. Line 399-404: first, the order of reference should be revised. Second, this is two references, should be separated.

8. Line 446-450, this reference has been deleted in the context.

Reviewer 2 ·

Basic reporting

no comment

Experimental design

no comment

Validity of the findings

no comment

Additional comments

The authors have reported the genetic map and sex-lined SNPs for Penaeus japonicus. The manuscript has been largely improved after the first round of review. However, there are also descriptions or analysis improper.
1. The validation of the sex-linked SNP is missing in Result.
2. The number of linakge group and the number of chromosome in karyotype analysis is still inconsistant. Even current genetic map is also of high quality and does not affect sex-linked SNP detection, it still can be improved. The authors has provided the relationship between LOD and number of linakge group in Figure S1. As the authors describe, the number of linakge group is 41 when the LOD set to 12. Check the Figure S1, when the LOD set to 17-21, the linage group would increae to 43, in which the number of liankge group is stabler. The threshold of site mumber in each linage group can be adjusted more according to the difference between the site number in the shortest linkage group and that in debris. Usually, the LOD can be selected according to the number of chromosome accroding to karyotype analysis and is less related with the threshold of site bumber for those data with hundreds and thousands of SNP, which almost cover the whole chromosomes. Thus, adjusting the LOD is encouraged.

---

## Round 0.3 · accepted · Accept

Thank you for providing sufficient rebuttal to the concerns raised by the reviewer and me.

Reviewer 2 ·

Basic reporting

none

Experimental design

none

Validity of the findings

none

Additional comments

According to the authors's speculation that the chromosome rearrangements would be an interesting points. However, it is doubtful two chromosomes rearranged in such a short time. Comparing chromosome rearrangements, problems with data analysis, maybe instrinsic to the species or the data, is more likely. This is still a high-quality paper, and the problem has no influence on sex marker develoopment.